# Hyperparameter Loss Surfaces Are Simple Near their Optima

**Nicholas Lourie**
New York University
nick.lourie@nyu.edu

**He He**
New York University
hhe@nyu.edu

**Kyunghyun Cho**
New York University & Genentech
kyunghyun.cho@nyu.edu

## Abstract

Hyperparameters greatly impact models' capabilities; however, modern models are too large for extensive search. Instead, researchers design recipes that train well across scales based on their understanding of the hyperparameters. Despite this importance, few tools exist for understanding the hyperparameter loss surface. We discover novel structure in it and propose a new theory yielding such tools. The loss surface is complex, but as you approach the optimum simple structure emerges. It becomes characterized by a few basic features, like its effective dimension and the best possible loss. To uncover this *asymptotic regime*, we develop a novel technique based on random search. Within this regime, the best scores from random search take on a new distribution we discover. Its parameters are exactly the features defining the loss surface in the asymptotic regime. From these features, we derive a new asymptotic law for random search that can explain and extrapolate its convergence. These new tools enable new analyses, such as confidence intervals for the best possible performance or determining the effective number of hyperparameters. We make these tools available at https://github.com/nicholaslourie/opda.

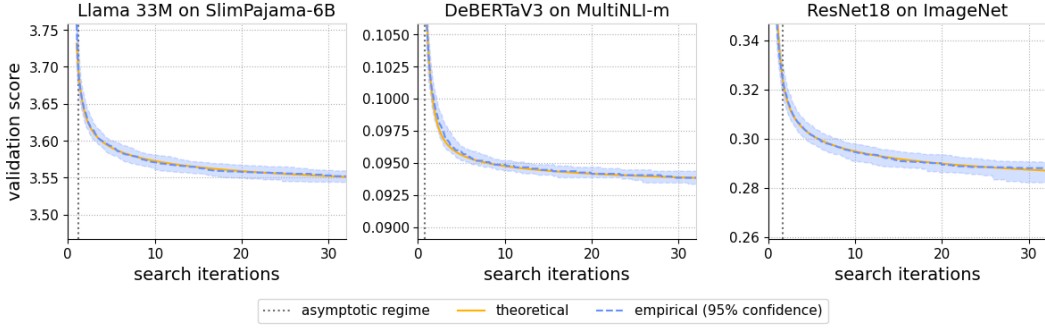

Figure 1: The hyperparameter loss surface has simple structure near the optimum. Using this structure, we can reason about how the validation score will improve as we run an algorithm like random search. The plots compare the *theoretical* functional form against the *empirical* rate of progress using 1,024 training runs in each. The ground truth *(dashed blue)* closely adheres to the theoretical form *(solid yellow)*, with that form remaining fully inside its 95% confidence bands. *Across all three scenarios—language model pretraining (log loss), supervised finetuning (error rate), and image classification (error rate)—the simple structure near the optimum drives the practical outcomes of hyperparameter search after just 1 or 2 iterations.*

## 1  Introduction

Hyperparameters affect every aspect of foundation models, from efficiency to generalization. Unfortunately, extensive hyperparameter search becomes impossible at scale. Instead, researchers must design recipes that work across scales based on their understanding of the hyperparameters. Still, few tools exist for understanding the hyperparameter loss surface.

We want to understand the surface, but it is too expensive to explore it—instead we must exploit its structure. We need an empirical theory of the hyperparameter loss surface. A similar theory, scaling laws, has had tremendous success at balancing data and compute without search (Rosenfeld et al., 2020; Kaplan et al., 2020; Hoffmann et al., 2022); however, scale is just one aspect of a model's design. Researchers must also consider things like the data mix, regularization, and architecture. μTransfer offers insights on optimization hyperparameters such as the initialization and the learning rate (Yang et al., 2021); nonetheless, it does not offer a general theory of the loss surface or reveal whether it has deeper structure.

We discover such structure exists and propose a new theory describing it. While the hyperparameter loss surface is complex, as you approach the optimum simple structure emerges. In a large area around it, the surface's structure is dominated by a few basic features: the effective dimension, the noise due to random seeds, and the best possible loss. Here, in the *asymptotic regime*, the surface looks like a quadratic polynomial with additive, normally distributed noise.

To find the asymptotic regime, we look for the shadow it casts on random search. In it, the scores from random search take on a specific distribution: the *noisy quadratic*. So, we look for a threshold after which the scores follow this distribution. Imagine you run random search, obtain validation scores, and plot their distribution, then you will see the distribution's tail matches a noisy quadratic. The point at which they match defines the asymptotic regime.

The noisy quadratic comes from a new limit theorem we prove for random search. Under regularity conditions, the best validation scores will follow a noisy quadratic. This theorem explains and extrapolates how random search converges—uncovering its asymptotic law.

Beyond search, the noisy quadratic captures properties of the hyperparameter loss surface. Each of the distribution's parameters corresponds to a different one: the effective number of hyperparameters, the variance due to random seeds, the concentration of probability near the optimum, and the best hyperparameters' loss. By fitting the distribution, we can estimate these properties. Thus, the noisy quadratic lets us find both where simple structure emerges and what it looks like. Better still, since the noisy quadratic is a parametric distribution, we can construct confidence intervals for these properties using maximum likelihood theory.

Of course, a good theory must reflect the data. We validate our theory in three practical scenarios: language model pretraining, supervised finetuning, and image classification (§4). Training 1,024 models in each, we test that our theoretical form fits the empirical distribution from random search (§4.1). In all three scenarios, the theoretical form adheres closely to the ground truth. Moreover, the asymptotic regime is always large—occupying a third to half of the search space. Beyond fit, we test our assumptions: is the noise actually normal with constant variance (§4.2)? In fact, it converges to normality long before the asymptotic regime, and while the variance begins inflated it quickly converges to a constant. Last, we test our theory's application. In each scenario, we extrapolate how random search will converge based on the first 48 search iterations (§4.3). While our point estimates mostly smooth their nonparametric baselines, the confidence bands show a dramatic improvement.

With our theory, researchers can identify the asymptotic regime, understand its structure, extrapolate how random search will converge, and infer properties of the hyperparameter loss surface—all while quantifying their uncertainty. So that others may use these tools in their own research, we make them available at: https://github.com/nicholaslourie/opda.

---

**Simple Structure Emerges Around the Optimum**

**Simple Structure.** In a large area around the optimum, the hyperparameter loss surface is approximately a quadratic polynomial plus noise which is normally distributed with constant variance.

**Around the optimum.** This structure emerges for all the points whose loss is better than some threshold. Beyond this threshold, the scores from random search follow a noisy quadratic distribution *(right)*.

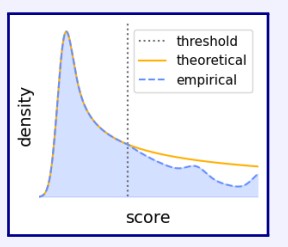

---

## 2 A Theory Based on Simple Structure

Our theory starts with a simple claim: near the best hyperparameters, the loss surface is approximately quadratic with additive normal noise. This structure emerges for those hyperparameters ($\boldsymbol{x}$) whose expected loss ($Y$) is better than some threshold: $\mathbb{E}[Y \mid \boldsymbol{x}] \leq \theta$, and that threshold defines the *asymptotic regime*. But, how do we find it?

Finding the asymptotic regime in a high dimensional search space would be challenging; instead, we look for the effect it leaves upon random search. Within the asymptotic regime, the scores found by random search follow a new parametric family: the noisy quadratic distribution. By finding where these scores converge to it, we find the threshold defining the asymptotic regime. The noisy quadratic's parameters then let us infer other properties of the loss surface, such as the best possible performance and its effective dimension.

Here, we derive these results informally, for proofs see §F.

### 2.1 Formalizing Hyperparameter Search

Imagine fitting a neural network—perhaps pretraining a language model.[1] Many choices must be made: Which architecture? What learning rate? Each one is a hyperparameter, and each hyperparameter takes a value from the *hyperparameter search space*, $\boldsymbol{x} = [x_1, \ldots, x_d] \in \mathbb{X}$. Evaluating the network then produces a score, $y \in \mathbb{Y} \subset \mathbb{R}$, such as cross-entropy. This score is a random variable that depends on the initialization and the data order as well as the hyperparameters: $Y \mid \boldsymbol{x}$. Its conditional distribution defines the *hyperparameter loss surface*, and in particular we will be interested in its mean: $g(\boldsymbol{x}) = \mathbb{E}[Y|\boldsymbol{x}]$.

To optimize it, we use a hyperparameter tuning algorithm. Each round, the algorithm (randomly) selects a hyperparameter configuration, $\boldsymbol{X}_i$, evaluates it to obtain a score, $Y_i$, then keeps the best one found so far:

$$T_k := \min_{i=1\ldots k} Y_i \tag{1}$$

We can think of $k \to T_k$ as a random function, which we call the *tuning process*.

The tuning process captures the whole distribution of outcomes from hyperparameter search; however, it is a complex, high dimensional object. Instead of the whole distribution, it is convenient to consider a summary like the median. Letting $\mathbb{M}[X]$ be the median of $X$, the *tuning curve* is the function, $\tau : \mathbb{R}_{>0} \to \mathbb{Y}$:[2]

$$\tau(k) := \mathbb{M}[T_k] \tag{2}$$

The tuning curve concisely describes how hyperparameter search might progress. In general, it depends on the model, the search space, and the search algorithm. Holding the model fixed, we can compare the efficiency of different tuning algorithms; holding the *algorithm* fixed, we can compare the difficulty of tuning different *models*.

A simple standard for comparing models is *random search*. In it, we sample hyperparameter configurations from a fixed *search distribution*, $\boldsymbol{X}_i \sim \mathcal{X}$. Since configurations are sampled independently, their scores are independent as well. In essence, the scores come from a fixed *score distribution*, $Y_i \sim \mathcal{Y}$. By analyzing this distribution, we can understand the whole tuning process. For example, the best loss after $k$ rounds is just the minimum of $k$ draws, and the minimum's cumulative distribution function (CDF), $F_k(y) = \mathbb{P}(T_k \leq y)$, has a simple relationship to the CDF of one sample, $F(y) = \mathbb{P}(Y_i \leq y)$:

$$F_k(y) = 1 - \mathbb{P}\left(\min_{i=1\ldots k} Y_i > y\right) = 1 - \prod_{i=1\ldots k} \mathbb{P}(Y_i > y) = 1 - (1 - F(y))^k \tag{3}$$

Thus, the distribution from one round of random search defines the distribution from $k$ rounds. If we can estimate the distribution's tail, then we can extrapolate infinitely into the future—capturing the entire tuning process. To accomplish this, we need to find an appropriate parametric form.

---

[1]This section closely follows our formalization in Lourie et al. (2024) (§3.1).

[2]Following Lourie et al. (2024), we extend the definition of the tuning curve from $k \in \mathbb{N}$ to all $k \in \mathbb{R}_{>0}$ by defining $T_k$ as the random variable with CDF $F(y)^k$.

Figure 2: With more iterations, random search finds better hyperparameters. As the best score improves, the region of better hyperparameters shrinks around the optimum; thereby, *the Taylor polynomial gives better approximations at the hyperparameters that improve the score.*

## 2.2 The Deterministic Case

Imagine the loss surface has no noise—that evaluating hyperparameters is deterministic. How would random search behave? At any point in time, the only hyperparameters that matter are the ones that might improve the loss. As search continues, you find better hyperparameters, and the region of even better ones shrinks about the optimum. As this region shrinks, the Taylor polynomial becomes a better approximation within it. Figure 2 illustrates this idea.

At the optimum, $(\boldsymbol{x}_*, y_*)$, the gradient is $\boldsymbol{0}$ so the Taylor series is given by the Hessian, $H_{\boldsymbol{x}_*}$:

$$g(\boldsymbol{x}) \approx y_* + \frac{1}{2}(\boldsymbol{x} - \boldsymbol{x}_*)^T H_{\boldsymbol{x}_*}(\boldsymbol{x} - \boldsymbol{x}_*) \tag{4}$$

At the same time, any continuous probability density is roughly constant on a small enough interval; thus, near the optimum, the search distribution is approximately uniform.

Putting these facts together, we derive the tail of the score distribution via a geometric argument. Consider the event $Y = g(\boldsymbol{X}) \leq y$. Rearranging the Taylor approximation gives:

$$\frac{1}{2}(\boldsymbol{x} - \boldsymbol{x}_*)^T H_{\boldsymbol{x}_*}(\boldsymbol{x} - \boldsymbol{x}_*) \leq y - y_* \tag{5}$$

Since $\boldsymbol{x}_*$ is a minimum, the Hessian is positive semi-definite. If we marginalize out the null dimensions, then this equation defines an ellipsoid. $\mathbb{P}(Y \leq y)$ is proportional to its volume, which is proportional to $(y - y_*)^{d_*/2}$, where $d_*$ is the rank of the Hessian. Thus, as $y \to y_*$:

$$F(y) = \mathbb{P}(Y \leq y) \propto (y - y_*)^{d_*/2} \tag{6}$$

Motivated by this analysis, we define the *convex quadratic distribution*, $\mathcal{Q}_{\min}(\omega, \beta, \gamma)$, by $F(y; \omega, \beta, \gamma) := \omega(y - \alpha)^{\gamma/2}$. Usually, we prefer an alternative parametrization. Let $\beta$ be the maximum of the distribution's support. $F(\beta) = \omega(\beta - \alpha)^{\gamma/2} = 1$, so $\omega = (\beta - \alpha)^{-\gamma/2}$ thus:

$$F(y; \alpha, \beta, \gamma) := \left(\frac{y - \alpha}{\beta - \alpha}\right)^{\gamma/2} \tag{7}$$

We can differentiate the CDF to obtain the probability density function (PDF):

$$f(y; \alpha, \beta, \gamma) = \frac{\gamma}{2(\beta - \alpha)}\left(\frac{y - \alpha}{\beta - \alpha}\right)^{\frac{\gamma-2}{2}} \tag{8}$$

Each parameter relates to a different aspect of the hyperparameter loss surface: $\alpha$ is *the best possible score*, $\beta$ measures how *concentrated* the distribution is near the minimum, and $\gamma$ is *the effective number of hyperparameters*—which is always less than the nominal number.

In summary, we derive a new parametric family: *the quadratic distribution*. This family describes the score distribution's tail when optimizing a deterministic function via random search. For *minimization*, the *left* tail approaches the *convex* quadratic distribution; for *maximization* the *right* tail approaches the *concave* quadratic distribution. We give formulas for both in §A. For more details on the derivation, see §E.

## 2.3 The Stochastic Case

The validation score is rarely a deterministic function of the hyperparameters. More often, the score varies due to random factors such as the initialization and data order.

Still, even when the scores are random, their conditional mean, $\mathbb{E}[Y|\mathbf{X}]$, is not. We could apply our previous analysis to the mean, but we would need to bridge the gap between it and the actual observations. Taking inspiration from classic regression analysis, we might consider whether $Y$ varies about the mean with additive noise. Let $g(\mathbf{X}) = \mathbb{E}[Y|\mathbf{X}]$, then we assume $Y = g(\mathbf{X}) + E$ where $E \sim \mathcal{N}(0, \sigma)$. This simple assumption seems too good to be true, but in fact it gives a great fit to the data (§4.1). Even more surprisingly, if you retrain the same hyperparameters many times, the scores do in fact become normally distributed with constant variance as you enter the asymptotic regime (§4.2). Thus, additive noise offers a realistic model for this random variation.

Assuming $Y = g(\mathbf{X}) + E$, the tail of $g(\mathbf{X})$ converges to a quadratic distribution, so:

$$Y \approx Q + E, \qquad Q \sim \mathcal{Q}_{\min}(\alpha, \beta, \gamma), \ E \sim \mathcal{N}(0, \sigma) \tag{9}$$

This sum defines a new family: the *noisy quadratic distribution*, $\mathcal{Q}(\alpha, \beta, \gamma, \sigma)$. It comes in two variants: the *convex* ($\mathcal{Q}_{\min}$) and *concave* ($\mathcal{Q}_{\max}$) noisy quadratic distributions. Moreover, when $\sigma = 0$, we recover the (noiseless) quadratic distribution as a special case.[3]

Let $\Phi$ be the CDF of the standard normal distribution. The CDF of the noisy quadratic is:

$$F(y; \alpha, \beta, \gamma, \sigma) = \Phi\left(\frac{y - \beta}{\sigma}\right) + \mathbb{E}_0^1\left[V^{\gamma/2}\right], \qquad V \sim \mathcal{N}\left(\frac{y - \alpha}{\beta - \alpha}, \frac{\sigma}{\beta - \alpha}\right) \tag{10}$$

The noisy quadratic's PDF is:

$$f(y; \alpha, \beta, \gamma, \sigma) = \frac{\gamma}{2(\beta - \alpha)}\mathbb{E}_0^1\left[V^{\frac{\gamma-2}{2}}\right], \qquad V \sim \mathcal{N}\left(\frac{y - \alpha}{\beta - \alpha}, \frac{\sigma}{\beta - \alpha}\right) \tag{11}$$

Thus, we can express the noisy quadratic distribution's CDF and PDF in terms of properties of the normal distribution. For mathematical details, see §F.2. Equations 10 and 11 require numerical methods to evaluate, so we provide robust implementations in our library opda.

In summary, we extend the quadratic distribution to a more general family: *the noisy quadratic distribution*. This family describes the score distribution—and thus the outcomes—of random search in typical deep learning scenarios. When *minimizing*, the score distribution's *left* tail converges to a *convex* noisy quadratic; when *maximizing*, its *right* tail converges to a *concave* noisy quadratic. See §B for formulas for both.

## 2.4 Applying the Theory in Practice

The noisy quadratic distribution is a powerful tool for studying neural networks. With it, we can answer two types of questions: we can use it to reason about random search, and we can use random search to understand the hyperparameter loss surface.

For random search: its convergence is described by the tuning curve, and the tuning curve is determined by the noisy quadratic. In particular, random search is fast when the effective number of hyperparameters ($\gamma$) is low.[4] For a given model, we can find how easy it is to tune by estimating its tuning curve using the noisy quadratic (e.g., see §4.3).

For the hyperparameter loss surface: its most important properties are captured by the noisy quadratic's parameters. When minimizing (maximizing), $\alpha$ ($\beta$) is the average score of the best hyperparameters and $\gamma$ is the effective number of them. We can estimate these quantities by fitting the noisy quadratic. Moreover, we can use maximum likelihood theory to test hypotheses and create confidence intervals for them (e.g., via a likelihood ratio test). For example, you could test if adding a new hyperparameter increases the effective number.

---

[3]When the variant is clear from context, we write the distribution unadorned: $\mathcal{Q}(\alpha, \beta, \gamma, \sigma)$. We differentiate the quadratic, $\mathcal{Q}(\alpha, \beta, \gamma)$, and noisy quadratic, $\mathcal{Q}(\alpha, \beta, \gamma, \sigma)$, by the presence of $\sigma$.

[4]In our derivation (§2.2), the effective number of hyperparameters is the rank of the Hessian.

To apply our theory, you must pick a search space and search distribution. A few tips. With a small number of discrete hyperparameters, you can evenly stratify over them to ensure good coverage. More generally, the noisy quadratic emerges when there exist coordinates in which the hyperparameters are uniform and the loss surface adheres to its 2nd order Taylor polynomial. Sampling on the right scales really helps. Start bounded hyperparameters on a logit, positive ones on a log, and real-valued ones on a linear scale. Afterwards, adjust as necessary. Sample uniformly (on that scale) between the bounds you would normally choose for a grid search. Tighter bounds speed up convergence, but they still must contain the optimum. To find the asymptotic regime, fit the noisy quadratic to the empirical CDF (eCDF) with several thresholds until you find the loosest one that gives a good visual fit.[5]

## 3 Experimental Setup

To test our theory, we run random search on representative scenarios: Llama 33M (Touvron et al., 2023) on SlimPajama (Soboleva et al., 2023) for language modeling, DeBERTaV3 (He et al., 2023) on MultiNLI (Williams et al., 2018) for supervised finetuning, and ResNet18 (He et al., 2016) on ImageNet (Russakovsky et al., 2015) for vision pretraining. To fully capture the score distribution, we train 1,024 separate hyperparameter configurations in each.

Llama 33M is a smaller variant of Llama, a causal transformer (Vaswani et al., 2017). We pretrain it on SlimPajama-6B,[6] a subset of the SlimPajama web corpus. We tune the learning rate, $\beta_1$ and $\beta_2$ for Adam (Kingma & Ba, 2015), warmup steps, weight decay, and dropout.

DeBERTaV3 is a pretrained BERT-like model (Devlin et al., 2019). In Lourie et al. (2024), we finetuned it on MultiNLI, a natural language inference benchmark. We tuned the learning rate, fraction of first epoch for warmup, batch size, number of epochs, and dropout.

ResNet18 is a convolutional network with residual connections. We train it with momentum SGD and a 1-cycle policy (Smith & Topin, 2018) on ImageNet, an image classification benchmark used for vision pretraining. We tune the learning rate, peak epoch, momentum, batch size, epochs, weight decay, label smoothing, and blurpool (an architectural parameter).

For the models' search distributions see §C. Our analysis and random search results (including sampled configurations and full learning curves) are available in opda (v0.8.0).

We validate our theory in three parts.

**Assessing Goodness of Fit.** We confirm the noisy quadratic matches the score distribution from random search. To do so, we estimate the score distribution using the eCDF and the nonparametric confidence bands from Lourie et al. (2024). We then fit the noisy quadratic to its tail via censored maximum spacing estimation (Cheng & Amin, 1983; Ranneby, 1984). We select the threshold for the asymptotic regime via visual diagnostics.

**Testing Additive Normal Errors.** We verify our strongest assumptions: normality and homoskedasticity of the variation due to random seeds. For this, we take the ResNet18 results, pick the configurations scoring at the 12.5th, 25th, up to 100th percentile, and retrain each 128 times to obtain large samples from score distributions with fixed hyperparameters.

**Estimating and Extrapolating the Tuning Curve.** To demonstrate how to use our results to evaluate a model, we subsample 48 iterations of random search without replacement from the full 1,024 for each model. For the ground truth eCDF, we use all 1,024 iterations. For nonparametric estimates, we construct the eCDF and Lourie et al.'s (2024) confidence bands from the subsample. For parametric estimates, we select the asymptotic regime via visual diagnostics using only the subsample, fit the noisy quadratic distribution to the tail via censored maximum spacing estimation (Cheng & Amin, 1983; Ranneby, 1984), and compute parametric confidence bands from the nonparametric ones as consonance regions (Easterling, 1976). We compute these via brute-force search with a grid of 64 log-spaced values for $\sigma$, 128 linearly spaced values for $\alpha$, and 256 linearly spaced values for $\beta$.

---

[5]For an example, see the Extrapolating Random Search section of the Validating the Parametric Analysis in Practice notebook in opda (v0.8.0).

[6]https://huggingface.co/datasets/DKYoon/SlimPajama-6B

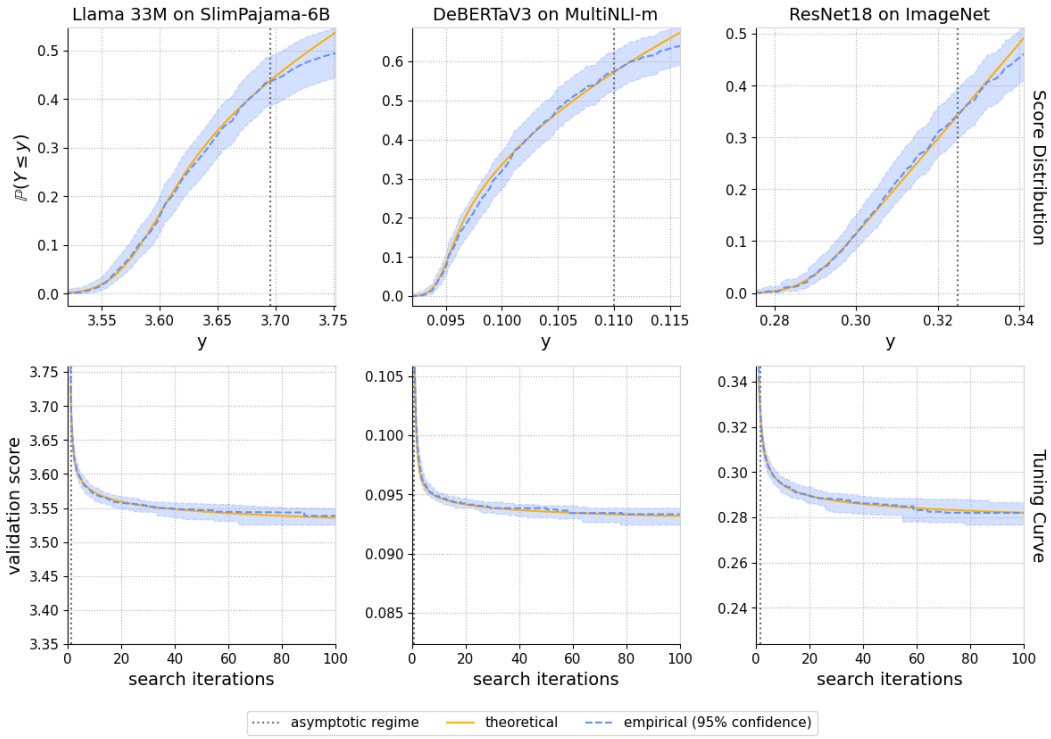

Figure 3: A comparison of the score distribution (*empirical*) and noisy quadratic (*theoretical*). The top row depicts CDFs, the bottom row depicts tuning curves. Each column corresponds to a different scenario: pretraining Llama 33M on SlimPajama-6B (cross-entropy), finetuning DeBERTaV3 on MultiNLI (error rate), and training ResNet18 on ImageNet (error rate). The *asymptotic regime* is the performance threshold beyond which the theoretical approximations apply. All estimates use the scenarios' full 1,024 iterations of random search. *In the asymptotic regime, the noisy quadratic distribution matches the score distribution from random search.*

## 4   Testing the Theory

### 4.1   Assessing Goodness of Fit

Our main claim is that simple structure in the hyperparameter loss surface determines practical outcomes from search. Using this structure, we derived theoretical forms for the score distribution and tuning curve. Figure 3 compares these forms to what you actually observe.

Across three scenarios, the empirical and theoretical distributions show an excellent fit. In each, both the noisy quadratic's CDF and its median tuning curve closely adhere to the ground truth. They remain within the 95% confidence bands at all times. And, as theory predicts, the point estimates fit the ground truth almost perfectly in the asymptotic regime.

Besides showing our assumptions are satisfied, these results show the theory is practically relevant. It is not enough for the hyperparameter loss surface to have structure at the optimum; that structure must occupy enough space around it to be useful. Figure 3 confirms this. In each scenario, a large fraction of the score distribution falls within the asymptotic regime: 44% for Llama 33M, 57% for DeBERTaV3, and 34% for ResNet18. These search spaces are broad, typical of what practitioners might use when tuning a new model. Still, almost the entire tuning curve falls in the asymptotic regime, from the first few iterations. These results also generalize when you use the same search space on new architectures (see §D). Empirically, the asymptotic regime explains much of the behavior you see in practice.

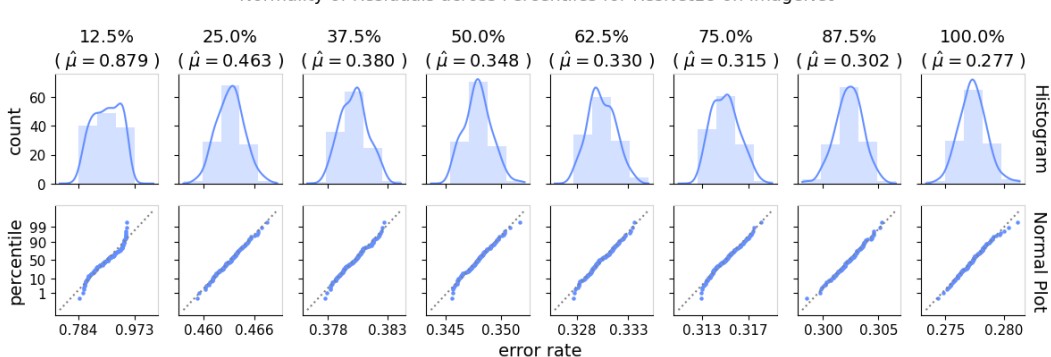

Figure 4: Diagnostics for the score distribution's normality given fixed hyperparameters. The top row shows histograms with kernel density estimates; the bottom shows Q-Q plots. Columns represent configurations across error rate percentiles for ResNet18 on ImageNet. *All except the worst performing hyperparameters demonstrate a very high degree of normality.*

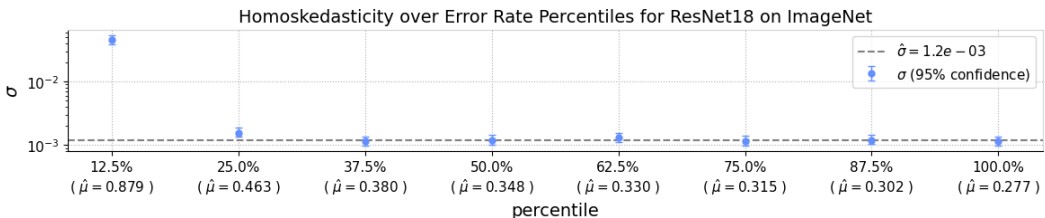

Figure 5: A comparison of standard deviations for the score given fixed hyperparameters. The *x*-axis gives configurations at different error rate percentiles for ResNet18 on ImageNet. Points are standard deviations at those percentiles. Confidence intervals are simultaneous. *The standard deviation quickly converges to a constant long before the asymptotic regime.*

### 4.2 Testing Additive Normal Errors

If you fix the hyperparameters, are the scores really normal with constant variance? Often, bad hyperparameters are unstable so we only expect this structure after the scores exceed some threshold (the asymptotic regime). We take search results from ResNet18, pick the configurations at the 12.5%, 25%, to 100% best error rates, and retrain each 128 times. The claim has two parts: the scores are normally distributed, and their variances are constant.

**Testing Normality.** We test normality using the venerable normal probability plot. In addition, we show histograms and kernel density estimates (KDEs) to offer a more intuitive visualization. Figure 4 displays the results. On the top, besides the worst hyperparameters, the distributions exhibit a familiar bell curve. The normal probability plots are even more decisive. On the bottom, the sample quantiles almost all fall on the $y = x$ line, corresponding to the quantiles of the normal. Each plot from 25% on up displays a high degree of normality.

**Testing Constant Variance.** We test constant variance by plotting simultaneous confidence intervals for the standard deviation across the error rates. As we have confirmed normality, we use the $\chi^2$ confidence interval for the standard deviation of a normal distribution; as the intervals are independent, we make them simultaneous via a Šidák correction. Figure 5 shows the result. The standard deviation drops to a constant around 37.5%. From then on, all confidence intervals contain a common value (1.2e-3, the mean of the last three points). The intervals are small, so it is unlikely any large differences exist. Thus, the standard deviation begins inflated, but converges to a constant long before the asymptotic regime.

Both analyses suggest the same conclusion: bad hyperparameters exhibit bad structure, but as hyperparameters improve—as you approach the optimum—simple structure emerges.

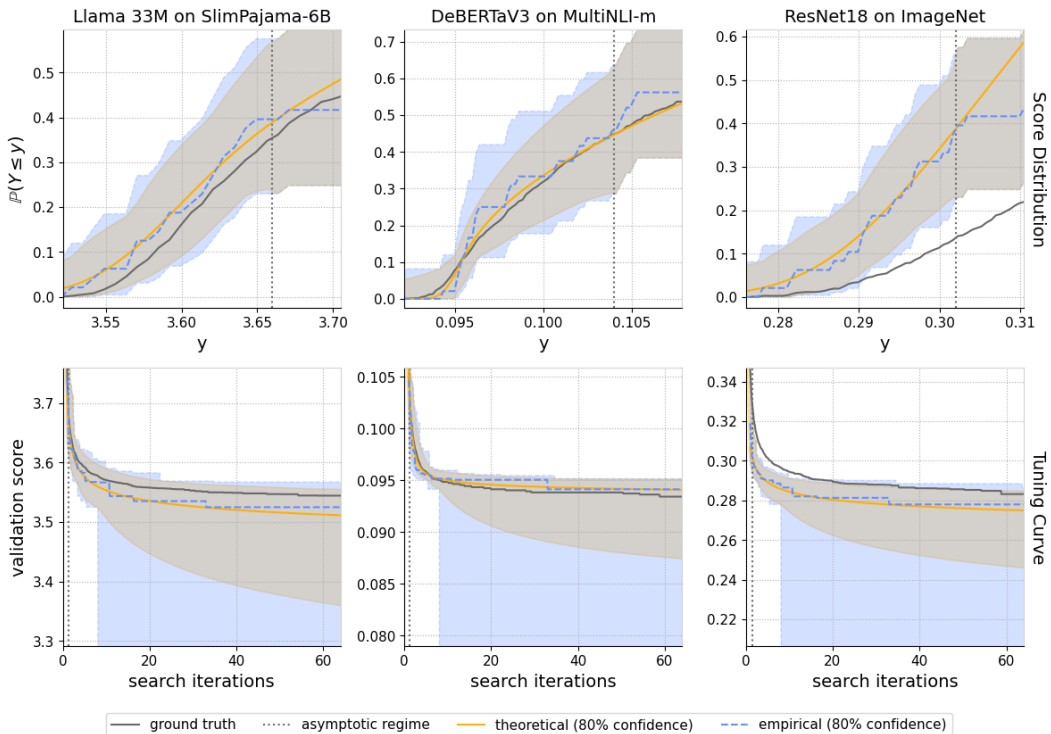

Figure 6: Examples of nonparametric (*empirical*) and noisy quadratic (*theoretical*) estimates. The top shows CDFs, the bottom shows tuning curves. Columns are different scenarios: pretraining Llama 33M on SlimPajama-6B (cross-entropy), finetuning DeBERTaV3 on MultiNLI (error rate), and training ResNet18 on ImageNet (error rate). The *asymptotic regime* is the threshold beyond which the noisy quadratic was fit. Estimates use 48 iterations of search. *The noisy quadratic distribution gives tighter bands for the tuning curve at the same confidence.*

## 4.3 Estimating and Extrapolating the Tuning Curve

We found simple structure in the hyperparameter loss surface. But, how can we use it? We want better tools for designing experiments—for answering questions like: did a new method actually improve the model, or did we just improve its hyperparameters? To answer this, we can estimate the model's *tuning curve*, or performance as a function of tuning effort (Dodge et al., 2019; Lourie et al., 2024); however, such estimates can be unreliable. With the noisy quadratic, we can construct better estimates and confidence bands. In particular, we can *extrapolate* how the score might improve as we continue hyperparameter search.

To explore this use case, we subsample 48 search iterations in each of the three scenarios. For each subsample, we plotted the eCDF to visually determine a threshold for the asymptotic regime. We chose thresholds based on where the eCDF began to show the expected structure. We estimated thresholds of 3.66 for Llama, 0.104 for DeBERTaV3, and 0.302 for ResNet18. Then, we fit the noisy quadratic distribution to these subsamples using the thresholds.

Figure 6 compares the parametric estimates against nonparametric baselines from Lourie et al. (2024). The parametric point estimates smooth out their nonparametric counterparts. This makes sense as both attempt to fit the same data: when the sample is not representative, they err in a similar way. This variation underscores the need for confidence bands. There, the approaches give dramatically different results. The parametric bands enclose the ground truth when the nonparametric ones do;[7] however, while the nonparametric bands become trivial after 8 iterations, the parametric bands extrapolate past all 48 used to construct them.

---

[7]At 80% confidence, at least 1 of the 3 bands will fail to contain the ground truth 48.8% of the time.

## 5 Related Work

Hyperparameters have always been an essential part of deep learning (Orr & Müller, 1998; Montavon et al., 2012); however, foundation models pose new challenges due to their cost.

Some researchers have sought theoretical solutions, such as $\mu$Transfer (Yang et al., 2021). $\mu$Transfer reparametrizes several important hyperparameters, such as the learning rate or initialization variance, so that their optimal values stay constant across scales. When successful, this lets you find hyperparameters at small scales and transfer to large ones.

To balance resources at scale, researchers turn to empirical scaling laws (Hestness et al., 2017; Rosenfeld et al., 2020; Kaplan et al., 2020; Hoffmann et al., 2022). Scaling laws predict how loss improves as more resources become available. The first scaling laws revealed that loss has a power law relationship with parameters and data (Rosenfeld et al., 2020; Kaplan et al., 2020). Since then, researchers have discovered scaling laws for many other quantities (Liu et al., 2024; Ludziejewski et al., 2024; Kumar et al., 2025). Like our work, scaling laws find structure in the loss surface—most often a power law, which is equivalent to linear structure on log scales. Unlike our work, scaling laws focus on specific inputs. Aside from scale, there are many hyperparameters that impact performance.

Several authors have explored structure in the hyperparameter loss surface (Pushak & Hoos, 2018). Pushak & Hoos (2022) look at AutoML pipelines and find that, while researchers typically use complex optimization algorithms, the loss surface is often unimodal or even convex. Conversely, Sohl-Dickstein (2024) discovers intricate, fractal-like structure at the boundary of where training succeeds for neural networks. In contrast to prior work, we focus exclusively on the hyperparameters around the optimum—the area of most practical interest. By focusing there, we are able to uncover the quadratic structure with normal noise, a structure that is far more regular than those previously found.

We uncover this structure via a limit theorem for random search. Similar limits are explored in *extreme value theory* (Coles, 2001; de Haan & Ferreira, 2006). For example, the Pickands-Balkema-De Haan theorem gives conditions under which the tail of a distribution converges to a generalized Pareto distribution (Pickands, 1975; Balkema & de Haan, 1974), which relates closely to the (noiseless) quadratic. Rather than the general theorems of extreme value theory, we analyze a specific mechanism in order to build an empirical theory of the hyperparameter loss surface.

## 6 Conclusion

The hyperparameter loss surface has simple structure near the optimum: it is approximately quadratic with additive normal noise. Surprisingly, this structure describes the surface quite well and holds over a large region about the optimum—up to 57% of the search space.

Using this structure, we derived a theory of random search in deep learning. We developed a parametric family for the score distribution's tail. This family comes in two forms: the quadratic in the deterministic case, and the noisy quadratic in the stochastic one.

The noisy quadratic distribution generalizes the first, and has four interpretable parameters:[8] $\alpha$, the average performance of the best possible hyperparameters, $\beta$, a measure of the probability in the asymptotic regime, $\gamma$, the effective number of hyperparameters, and $\sigma$, the scale of the noise due to random seeds. These parameters correspond to characteristics of the loss surface. Thus, our theory lets you reason about random search based on properties of the loss surface, but also *to reason about the loss surface based on the behavior of random search*.

We studied the loss surface to design better tools for deep learning experiments. While hyperparameter tuning is an orthogonal goal, our discoveries might suggest more efficient algorithms, e.g. Bayesian optimization kernels that exploit the quadratic-normal structure.

Our theory offers new tools for deep learning and foundation model research. So that others may use them, we make them available at: `https://github.com/nicholaslourie/opda`.

---

[8]When maximizing instead of minimizing, the roles of $\alpha$ and $\beta$ are reversed.

## Acknowledgments

We thank the anonymous reviewers for their valuable feedback. In addition, we thank Wanmo Kang for helpful suggestions while developing this work. This work was supported by the Institute of Information & Communications Technology Planning & Evaluation (IITP) with a grant funded by the Ministry of Science and ICT (MSIT) of the Republic of Korea in connection with the Global AI Frontier Lab International Collaborative Research. This work was also supported by the Samsung Advanced Institute of Technology (under the project Next Generation Deep Learning: From Pattern Recognition to AI) and the National Science Foundation (under NSF Award 1922658). This work was supported in part through the NYU IT High Performance Computing resources, services, and staff expertise.

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

## A  The Quadratic Distribution

As derived in §2.2, the formulas for the *convex quadratic distribution* are:

$$F(y; \alpha, \beta, \gamma) := \left(\frac{y - \alpha}{\beta - \alpha}\right)^{\gamma/2} \tag{12}$$

$$f(y; \alpha, \beta, \gamma) = \frac{\gamma}{2(\beta - \alpha)} \left(\frac{y - \alpha}{\beta - \alpha}\right)^{\frac{\gamma - 2}{2}} \tag{13}$$

The equivalent formulas for the *concave quadratic distribution* are:

$$F(y; \alpha, \beta, \gamma) := 1 - \left(\frac{\beta - y}{\beta - \alpha}\right)^{\gamma/2} \tag{14}$$

$$f(y; \alpha, \beta, \gamma) = \frac{\gamma}{2(\beta - \alpha)} \left(\frac{\beta - y}{\beta - \alpha}\right)^{\frac{\gamma - 2}{2}} \tag{15}$$

The quadratic distribution is supported only on the interval $\alpha \leq y \leq \beta$. These formulas are valid within that interval. Outside of it, the density is 0; below it, the CDF is 0; above it, the CDF is 1. The quadratic distribution family is a special case of the four parameter beta distribution, and closely relates to its other special case, the power function distribution.

For more derivations, properties, and discussion of the quadratic distribution, see The Quadratic Distribution section of the Parametric Analysis notebook in opda (v0.8.0).

## B  The Noisy Quadratic Distribution

As derived in §2.3, the formulas for the *convex noisy quadratic distribution* are:

$$F(y; \alpha, \beta, \gamma, \sigma) = \Phi\left(\frac{y - \beta}{\sigma}\right) + \mathbb{E}_0^1\left[V^{\gamma/2}\right], \qquad V \sim \mathcal{N}\left(\frac{y - \alpha}{\beta - \alpha}, \frac{\sigma}{\beta - \alpha}\right) \tag{16}$$

$$f(y; \alpha, \beta, \gamma, \sigma) = \frac{\gamma}{2(\beta - \alpha)} \mathbb{E}_0^1\left[V^{\frac{\gamma - 2}{2}}\right], \qquad V \sim \mathcal{N}\left(\frac{y - \alpha}{\beta - \alpha}, \frac{\sigma}{\beta - \alpha}\right) \tag{17}$$

The equivalent formulas for the *concave noisy quadratic distribution* are:

$$F(y; \alpha, \beta, \gamma, \sigma) = \Phi\left(\frac{y - \alpha}{\sigma}\right) - \mathbb{E}_0^1\left[V^{\gamma/2}\right], \qquad V \sim \mathcal{N}\left(\frac{\beta - y}{\beta - \alpha}, \frac{\sigma}{\beta - \alpha}\right) \tag{18}$$

$$f(y; \alpha, \beta, \gamma, \sigma) = \frac{\gamma}{2(\beta - \alpha)} \mathbb{E}_0^1\left[V^{\frac{\gamma - 2}{2}}\right], \qquad V \sim \mathcal{N}\left(\frac{\beta - y}{\beta - \alpha}, \frac{\sigma}{\beta - \alpha}\right) \tag{19}$$

Unlike the quadratic distribution, the noisy quadratic is supported on the entire real line.

For more derivations, properties, and discussion of the noisy quadratic distribution, see The Noisy Quadratic Distribution section of the Parametric Analysis notebook in opda (v0.8.0).

## C  Models and Search Distributions

As described in §3, we use random search results for Llama 33M, DeBERTaV3, and ResNet18.

For Llama 33M, we used the following search distribution:

$$\mathtt{lr} \sim \mathrm{LogUniform}(1e{-}5, 1e{-}1)$$
$$\mathtt{beta1} \sim \mathrm{Uniform}(0.7, 1)$$
$$\mathtt{beta2} \sim \mathrm{Uniform}(0.8, 1)$$
$$\mathtt{warmup\_steps} \sim \mathrm{DiscreteUniform}(0, 3000)$$
$$\mathtt{weight\_decay} \sim \mathrm{LogUniform}(1e{-}4, 1e0)$$
$$\mathtt{dropout} \sim \mathrm{Uniform}(0, 0.1)$$

For DeBERTaV3, we used the following search distribution in Lourie et al. (2024):

$$\mathtt{batch\_size} \sim \mathrm{DiscreteUniform}(16, 64)$$
$$\mathtt{num\_epochs} \sim \mathrm{DiscreteUniform}(1, 4)$$
$$\mathtt{warmup\_proportion} \sim \mathrm{Uniform}(0, 0.6)$$
$$\mathtt{learning\_rate} \sim \mathrm{LogUniform}(1e{-}6, 1e{-}3)$$
$$\mathtt{dropout} \sim \mathrm{Uniform}(0, 0.3)$$

Note that `warmup_proportion` is the proportion of the first epoch only.

For ResNet18, we used the following search distribution:

$$\mathtt{epochs} \sim \mathrm{DiscreteUniform}(20, 100)$$
$$\mathtt{batch\_size} \sim \mathrm{DiscreteUniform}(\{128, 256, 512, 1024\})$$
$$\mathtt{lr} \sim \mathrm{LogUniform}(5e{-}3, 5e1)$$
$$\mathtt{proportion} \sim \mathrm{Uniform}(0, 0.8)$$
$$\mathtt{lr\_peak\_epoch} = \lfloor \mathtt{proportion} \times \mathtt{epochs} \rfloor$$
$$\mathtt{momentum} \sim \mathrm{Uniform}(0.7, 1.0)$$
$$\mathtt{weight\_decay} \sim \mathrm{LogUniform}(1e{-}6, 1e{-}3)$$
$$\mathtt{label\_smoothing} \sim \mathrm{Uniform}(0.0, 0.5)$$
$$\mathtt{use\_blurpool} \sim \mathrm{DiscreteUniform}(0, 1)$$

In §D, we present additional results that validate our theory's generality and how it applies across architectures. For it, we compare DeBERTaV3 to DeBERTa (He et al., 2021), both tuned using DeBERTaV3's search distribution above. We also run random search on AlexNet (Krizhevsky et al., 2012; Krizhevsky, 2014), ResNet18 (He et al., 2016), and ConvNext Tiny (Liu et al., 2022) using ResNet18's search distribution above, except fixing `use_blurpool` to 0 because ConvNext does not use maxpool (or blurpool) layers and thus we can not consistently apply the hyperparameter to all three.

## D  Generalization Across Architectures

While our theory is general, it is also asymptotic; thus, it is natural to ask: how quickly does the asymptotic approximation apply in practice? For Llama 33M, DeBERTaV3, and ResNet18 we saw the asymptotic regime covered 44%, 57%, and 34% of the score distribution—applying from the first or second iteration of random search. Still, perhaps the asymptotic

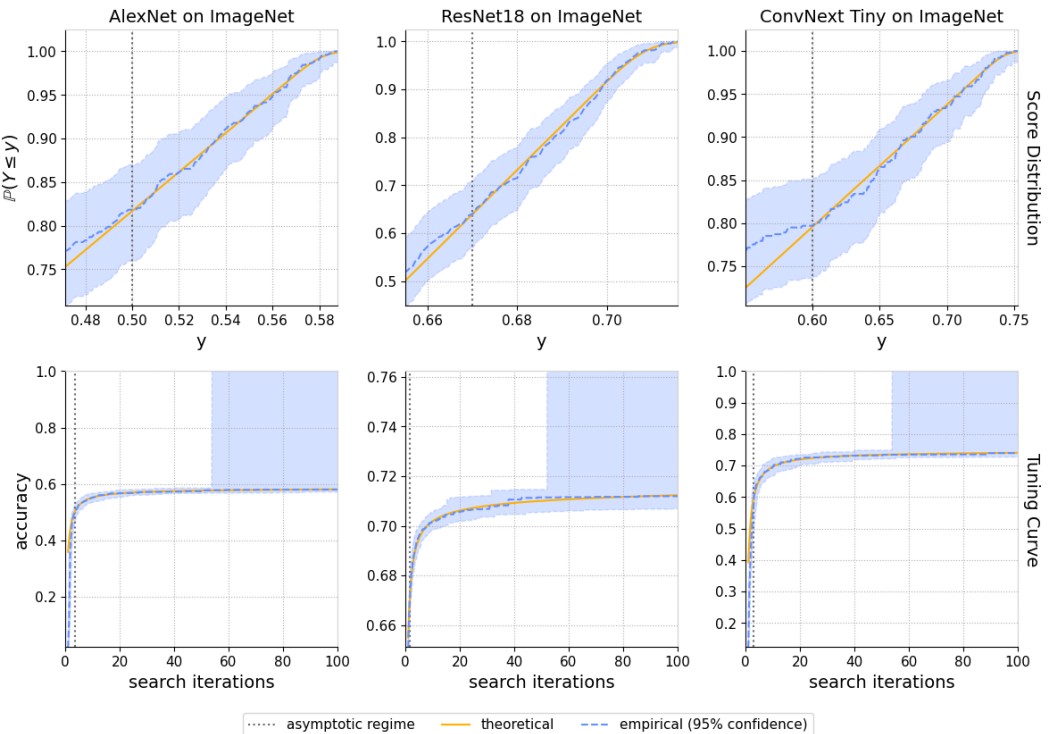

Figure 7: A comparison of the noisy quadratic (*theoretical*) and the score distribution (*empirical*) across different architectures trained on ImageNet. Each column corresponds to a different model: AlexNet, ResNet18, and ConvNext. All models use the same search distribution, and the estimates use 512, 495, and 512 iterations of random search, respectively. Empirical estimates are from the empirical distribution, while theoretical estimates use the noisy quadratic fitted to the tail via censored maximum spacing estimation. *For all three models tuned with the same search distribution, the asymptotic regime covers a large fraction of the search space and the noisy quadratic demonstrates an excellent fit to the score distribution.*

regime applies only because these architectures are so advanced, or the search spaces match them particularly well.

To investigate such questions, we compare ResNet18 with two other architectures: AlexNet (Krizhevsky et al., 2012; Krizhevsky, 2014) and ConvNext Tiny (Liu et al., 2022). AlexNet goes from ResNet into the past: many consider it the first major architecture of the current deep learning renaissance and, as such, it is considerably less advanced than ResNet—missing later innovations such as batch normalization or residual connections. On the other hand, ConvNext goes from ResNet into the future: it starts with the ResNet architecture and applies lessons learned from transformer-based models. We obtain 512, 495, and 512 iterations of random search for AlexNet, ResNet18, and ConvNext, using the same search distribution across all three (see §C). By using the same search distribution across all three models, we guarantee it is not unusually well-suited to any specific one.

Similarly, we use the same search space across DeBERTa (He et al., 2021) and DeBERTaV3, for which we obtain 1,024 search iterations each. Figures 7 and 8 present the results.

**The asymptotic regime is large in practice.** Across all architectures, the asymptotic regime is more than large enough to be practically relevant. Of the search distributions, it covers 18% for AlexNet, 36% for ResNet18,[9] and 20% for ConvNext Tiny as well as 54% for DeBERTa and 57% for DeBERTaV3. In other words, it characterizes the tuning curve after 1-4 iterations

---

[9]In this experiment use_blurpool is fixed to 0, which changes the asymptotic regime for ResNet18.

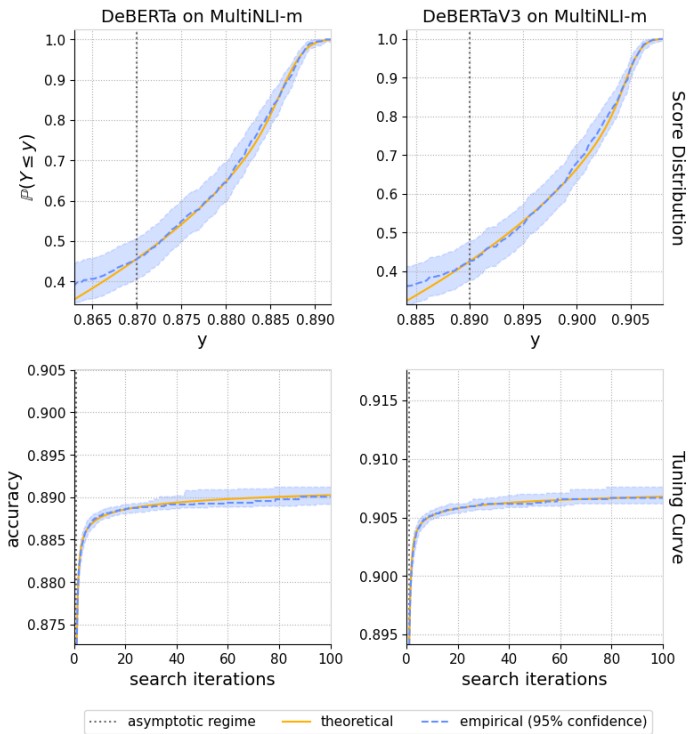

Figure 8: A comparison of the noisy quadratic (*theoretical*) and the score distribution (*empirical*) across different models finetuned on MultiNLI. Each column corresponds to a different one: DeBERTa and DeBERTaV3. Both models use the same search distribution, and the estimates each use 1,024 iterations of random search. Empirical estimates are from the empirical distribution, while theoretical estimates use the noisy quadratic fitted to the tail via censored maximum spacing estimation. *For both models tuned with the same search distribution, the asymptotic regime covers a large fraction of the search space and the noisy quadratic demonstrates an excellent fit to the score distribution.*

of random search. Thus, our theory describes random search with a realistic budget. The search distribution can not be the driving factor behind this result because we use the same one across each set of architectures. Moreover, while the better architectures display larger asymptotic regimes (e.g., ResNet18 and ConvNext), our theory even describes an older less advanced architecture like AlexNet after just a handful of search iterations.

**The effective number of hyperparameters is stable across architectures.** An interesting thing happens when we use the same search space across the different architectures: the effective number of hyperparameters ($\gamma$) remains constant. For AlexNet, ResNet, and ConvNext Tiny, the estimate of $\gamma$ is 2. DeBERTa and DeBERTaV3 exhibit a similar phenomenon: both models have $\gamma = 1$. This result suggests an intuitive conclusion: the effective number of hyperparameters seems to be more a property of the search space, i.e. the hyperparameters themselves. Thus, it exhibits some stability across models.

**Convergence is not necessary.** In modern deep learning, training is often limited by compute. As a result, our theory must apply even when the network is not trained to convergence. Fortunately, the ConvNext Tiny results demonstrate this to be true. Despite its name, ConvNext Tiny is significantly larger than ResNet18 (29M vs 12M parameters)—instead, it is more comparable to ResNet50. As our training recipe was chosen for ResNet18, it does not use enough compute (epochs) for ConvNext Tiny to fully converge. This fact is evident in the best accuracy achieved: 74.9% as opposed to 82.1% in Liu et al. (2022). Even in this compute-limited regime, the theory still obtains an excellent fit.

# E   Deriving the Quadratic Distribution

Recall §2.2, we wish to derive a limiting form for the score distribution's tail. Near the minimum, we approximate the hyperparameter loss surface, $g(\boldsymbol{x})$, by a Taylor polynomial and we approximate the search distribution by a uniform.

At the minimum, $(\boldsymbol{x}_*, y_*)$, the gradient is $\boldsymbol{0}$ and the Hessian is positive semi-definite with eigenvalues $\lambda_i \geq 0$. Without loss of generality, let the nonzero eigenvalues be $\lambda_1, \ldots, \lambda_{d_*}$. Since the Hessian is real and symmetric, it is diagonalizable: $H_{\boldsymbol{x}_*} = Q \Lambda Q^T$, where $Q$ is orthogonal and $\Lambda$ is the diagonal matrix of eigenvalues.

The second order Taylor polynomial can be written as:

$$
\begin{aligned}
g(\boldsymbol{x}) &\approx y_* + \frac{1}{2}(\boldsymbol{x} - \boldsymbol{x}_*)^T H_{\boldsymbol{x}_*}(\boldsymbol{x} - \boldsymbol{x}_*) \\
&= y_* + \frac{1}{2}(\boldsymbol{x} - \boldsymbol{x}_*)^T Q \Lambda Q^T (\boldsymbol{x} - \boldsymbol{x}_*) \\
&= y_* + \frac{1}{2}\left(\boldsymbol{x}' - \boldsymbol{x}'_*\right)^T \Lambda \left(\boldsymbol{x}' - \boldsymbol{x}'_*\right) \\
&= y_* + \frac{1}{2}\sum_{j=1}^{d_*} \lambda_i \left(x'_j - x'_{*j}\right)^2
\end{aligned}
$$

where $\boldsymbol{x}' = Q^T \boldsymbol{x}$, and we think of $Q^T$ as a change of coordinates. Since $Q^T$ is orthogonal, if $\boldsymbol{X}$ is (approximately) uniform then so is $\boldsymbol{X}' = Q^T \boldsymbol{X}$.

Consider the event $Y = g(\boldsymbol{X}) \leq y$. Rearranging the Taylor approximation, we obtain:

$$
\sum_{j=1}^{d_*} \frac{\lambda_i}{2}\left(\frac{x'_j - x'_{*j}}{\sqrt{y - y_*}}\right)^2 \leq 1
$$

This formula is the equation for an ellipsoid along the nonzero eigenvectors. Effectively, we marginalize over the null eigenvectors, along which we assume the loss surface is approximately constant.

The volume of this ellipsoid is proportional to $\sqrt{y - y_*}$ raised to the dimension, or $(y - y_*)^{d_*/2}$. Since the search distribution is approximately uniform, the probability of the event $Y = g(\boldsymbol{X}) \leq y$ is then proportional to this volume:

$$
F(y) = \mathbb{P}(Y \leq y) \propto (y - y_*)^{d_*/2}
$$

For an even more detailed discussion, simulations, and visualizations of the convergence, see the Approximating the Tail section of the Parametric Analysis notebook in opda (v0.8.0).

# F   Proofs & Theorems

In §2 and §E, we derived our results without emphasizing formality. We did this for two reasons. First, there are many ways to formalize the theorem—without a particular goal, any specific choice is arbitrary. Second, whether the limit applies in practice is ultimately an empirical question. Consider the normal distribution: numerous versions of the central limit theorem exist, each applying in its own context. What is important is not that one set of conditions produces the normal distribution, but that many do. Therefore, we expect it might appear and, accordingly, use diagnostics like normal probability plots to determine if it has. That in mind, we now illustrate one way to formalize things.

## F.1   The Deterministic Case

We prove a limit theorem for minimization via random search in the deterministic case.

First, we need the following proposition, which gives a kind of inverse continuity near the minimum:

**Proposition F.1.** *Let $\mathbb{X} \subset \mathbb{R}^d$ be compact, $\mathbb{Y} \subset \mathbb{R}$, $g : \mathbb{X} \to \mathbb{Y}$ continuous, and $y_* = g(\boldsymbol{x}_*)$ its unique minimum. Then $\forall \delta > 0$, $\exists \epsilon$ such that $|g(\boldsymbol{x}) - y_*| < \epsilon$ implies $\|\boldsymbol{x} - \boldsymbol{x}_*\| < \delta$.*

*Proof.* For contradiction, assume $\delta > 0$ is such that the conclusion is false. Let $\epsilon_i$ be any sequence such that $\epsilon_i \to 0$. For each $\epsilon_i$, there exists some $\boldsymbol{x}_i$ such that $|g(\boldsymbol{x}_i) - y_*| < \epsilon_i$ but $\|\boldsymbol{x}_i - \boldsymbol{x}_*\| > \delta$, otherwise the conclusion would be true.

Consider the sequence $\boldsymbol{x}_i$. Since $\mathbb{X}$ is compact, it has a convergent subsequence: $\boldsymbol{x}_{i_k} \to \boldsymbol{x}_\infty$. By construction, $|g(\boldsymbol{x}_{i_k}) - y_*| < \epsilon_{i_k}$. As $\epsilon_i \to 0$, we have $g(\boldsymbol{x}_{i_k}) \to y_*$, and because $g$ is continuous:

$$g(\boldsymbol{x}_\infty) = g\left(\lim_{i_k \to \infty} \boldsymbol{x}_{i_k}\right) = \lim_{i_k \to \infty} g(\boldsymbol{x}_{i_k}) = y_*$$

However, $\|\boldsymbol{x}_{i_k} - \boldsymbol{x}_*\| > \delta$ so $\boldsymbol{x}_\infty \neq \boldsymbol{x}_*$, contradicting uniqueness of the minimum. $\square$

**Theorem F.2.** *Let $\mathbb{X} \subset \mathbb{R}^d$ be compact, $\mathbb{Y} \subset \mathbb{R}$, $g : \mathbb{X} \to \mathbb{Y}$ thrice continuously differentiable, $y_* = g(\boldsymbol{x}_*)$ its unique minimum in the interior of $\mathbb{X}$, $H_{\boldsymbol{x}_*}$ the Hessian at $\boldsymbol{x}_*$ having full rank, and $\boldsymbol{X} \sim \mathcal{X}$ a distribution over $\mathbb{X}$ with continuous PDF, $\mu(\boldsymbol{x})$. If $Y = g(\boldsymbol{X})$ is a random variable with CDF $F(y)$, there exists a quadratic distribution with CDF $Q(y)$ such that $\lim_{y \to y_*} F(y)/Q(y) = 1$.*

*Proof.* Write the 2nd order Taylor approximation of $g$ at $\boldsymbol{x}_*$ as $t(\boldsymbol{x}) = y_* + 1/2(\boldsymbol{x} - \boldsymbol{x}_*)^T H_{\boldsymbol{x}_*}(\boldsymbol{x} - \boldsymbol{x}_*)$. Consider some neighborhood of $\|\boldsymbol{x} - \boldsymbol{x}_*\| < \delta$. By Proposition F.1, we can require $y$ be sufficiently close to $y_*$ to guarantee $\boldsymbol{x}$ is in it. Throughout the neighborhood, let $\epsilon$ be the Taylor approximation's worst case error:

$$t(\boldsymbol{x}) - \epsilon < g(\boldsymbol{x}) < t(\boldsymbol{x}) + \epsilon \tag{20}$$

Consider $F(y) = \mathbb{P}(Y \leq y)$. By Equation 20, $\mathbb{P}(t(\boldsymbol{x}) + \epsilon \leq y) \leq \mathbb{P}(g(\boldsymbol{x}) \leq y) \leq \mathbb{P}(t(\boldsymbol{x}) - \epsilon \leq y)$. We can write this equivalently as:

$$\mathbb{P}(t(\boldsymbol{x}) \leq y - \epsilon_1) \leq \mathbb{P}(g(\boldsymbol{x}) \leq y) \leq \mathbb{P}(t(\boldsymbol{x}) \leq y + \epsilon_1) \tag{21}$$

Let us analyze $\mathbb{P}(t(\boldsymbol{x}) \leq y)$.

We will need the fact that $\mathcal{X}$ is approximately uniform near $\boldsymbol{x}_*$. Let $c = \mu(\boldsymbol{x}_*)$. As $\mu$ is continuous, $\mu(\boldsymbol{x}) \to c$ as $\boldsymbol{x} \to \boldsymbol{x}_*$. Let $\eta$ be the maximum difference in the neighborhood:

$$c - \eta < \mu(\boldsymbol{x}) < c + \eta \tag{22}$$

In this sense, we can think of $\mathcal{X}$ as approximately uniform with density between $c \pm \eta$.

Returning to the Taylor approximation, $g$ is thrice continuously differentiable so the Hessian is real symmetric thus diagonalizable: $H_{\boldsymbol{x}_*} = U^T \Lambda U$, with $U$ an orthonormal matrix and $\Lambda = \text{diag}(\lambda_1, \ldots, \lambda_d)$ the eigenvalues. Think of $U$ as a change of coordinates, $\boldsymbol{u} = U\boldsymbol{x}$. Since $U$ is orthonormal with $|\det U| = 1$, by the change of variables theorem the density of $\mathcal{X}$ in these new coordinates is still approximately $c \pm \eta$.

Finally, consider the event: $\mathbb{P}(t(\boldsymbol{x}) \leq y)$. In the coordinates $\boldsymbol{u}$, $H_{\boldsymbol{x}_*}$ is a diagonal matrix and $t(\boldsymbol{u}) = y_* + 1/2 \sum_{i=1}^d \lambda_i (u_i - u_{*i})^2$; therefore, $t(\boldsymbol{u}) \leq y$ defines an ellipse:

$$\sum_{i=1}^d \frac{\lambda_i}{2} (u_i - u_{*i})^2 \leq y - y_*$$

The volume of this ellipse is:

$$(y - y_*)^{d/2} \left( \frac{\pi^{d/2}}{\Gamma\left(\frac{d}{2} + 1\right)} \prod_{i=1}^d \sqrt{\frac{2}{\lambda_i}} \right)$$

Take all the terms that do not depend on $y$ as a constant, $C$. The volume is then: $C(y - y_*)^{d/2}$. The probability $\mathbb{P}(t(\boldsymbol{x}) \leq y)$ is the density integrated over this volume. The density is

between $c - \eta$ and $c + \eta$, thus the probability is between products of these values and the volume:

$$C(y - y_*)^{d/2}(c - \eta) < \mathbb{P}(t(\boldsymbol{x}) \le y) < C(y - y_*)^{d/2}(c + \eta) \tag{23}$$

Combining Equations 21 and 23, we have:

$$C(y - \epsilon - y_*)^{d/2}(c - \eta) < \mathbb{P}(g(\boldsymbol{x}) \le y) < C(y + \epsilon - y_*)^{d/2}(c + \eta)$$

Using the parametrization of the (convex) quadratic distribution's CDF as $Q(y) = \omega(y - \alpha)^{\gamma/2}$, let $\omega = Cc$, $\alpha = y_*$, and $\gamma = d$. Then dividing by $Q(y)$ we have:

$$\frac{(c - \eta)}{c}\left(1 - \frac{\epsilon}{y - y_*}\right)^{d/2} < \frac{F(y)}{Q(y)} < \frac{(c + \eta)}{c}\left(1 + \frac{\epsilon}{y - y_*}\right)^{d/2} \tag{24}$$

Consider what happens as $y \to y_*$. By Proposition F.1 the neighborhood about $\boldsymbol{x}_*$ shrinks. As a result, $\eta \to 0$ and since $g$ is thrice differentiable the Taylor approximation's error goes to 0 at 3rd order while $y - y_*$ goes to 0 at 2nd order, thus $\epsilon/(y - y_*) \to 0$. Therefore, the upper and lower bounds in Equation 24 go to 1 and thus $F(y)/Q(y) \to 1$ as well. In other words:

$$\lim_{y \to y_*} \frac{F(y)}{Q(y)} = 1$$

$\square$

Thus, we obtain a limit theorem for random search under minimization, maximization being similar.

A few remarks are in order. We have shown convergence under one set of conditions; however, convergence can happen under other conditions as well. For example, we used uniqueness of the minimum to ensure that as $y$ approaches $y_*$, the corresponding $\boldsymbol{x}$ also approaches $\boldsymbol{x}_*$, the center of our Taylor approximation. If a finite number of distinct minima exist, this condition still holds as we approach the global minimum. Even with multiple global minima, they can be added together without issue.[10] For example, the volume of their ellipses will be: $\sum_{j=1}^{n} C_j(y - y_*)^{d/2} = (y - y_*)^{d/2}\sum_{j=1}^{n} C_j$. As this example shows, many variants of the theorem are possible.

One assumption in particular merits deeper discussion: that the Hessian is full rank. Empirically, this assumption is rarely true. In all our experiments, the effective number of hyperparameters was fewer than the nominal number—in other words, the Hessian was rank deficient. Here is one way to close this gap: if $g$ is constant along the kernel of the Hessian, then you can marginalize over the kernel and consider $g$ as a function of the quotient space, in which the Hessian will have full rank.

In the end, we just need the hyperparameter loss to be approximately quadratic in some coordinates for which the search distribution is approximately uniform. Designing the search space so these assumptions are better satisfied will speed up convergence. For example, you can search for each hyperparameter using a uniform distribution on the appropriate scale (e.g., a log scale for the learning rate). Similarly, you can tighten the search space around the optimum so the Taylor approximation is a better fit.

### F.2 The Stochastic Case

For the stochastic case, the noisy quadratic distribution is defined as the sum of a quadratic and a normal random variable. If the conditional mean $g(\boldsymbol{X}) = \mathbb{E}[Y|\boldsymbol{X}]$ satisfies the conditions of Theorem F.2, then it will converge to a quadratic distribution. If in addition $Y = g(\boldsymbol{X}) + E$, $E \sim \mathcal{N}(0, \sigma)$ then one just needs $\sigma$ to be small enough, otherwise the noise ($E$) will contaminate points where the quadratic distribution is a good approximation with the points where it is a bad one.

We derive the formulas for the CDF and PDF. We will show them for minimization.

---

[10]This reasoning also applies to discrete hyperparameters, which essentially multiply local minima.

First, we will need the definition of the *partial expectation from a to b*:

$$\mathbb{E}_a^b[Z] := \mathbb{P}(a \leq Z \leq b) \, \mathbb{E}[Z | a \leq Z \leq b] = \int_a^b z f_Z(z) dz \tag{25}$$

Now, we will prove the formulas.

**Proposition F.3.** *Let $Y = Q + E$, with $Q \sim \mathcal{Q}_{\min}(\alpha, \beta, \gamma)$ and $E \sim \mathcal{N}(0, \sigma)$. If $F_Y(y)$ is the CDF of $Y$ then:*

$$F_Y(y) = \Phi\left(\frac{y-\beta}{\sigma}\right) + \mathbb{E}_0^1\left[V^{\gamma/2}\right], \qquad V \sim \mathcal{N}\left(\frac{y-\alpha}{\beta-\alpha}, \frac{\sigma}{\beta-\alpha}\right)$$

*Proof.* Let $F_Q(y)$ denote the CDF of $Q$. Since $Y$ is the sum of two independent random variables, we can apply the convolution formula for the CDF of a sum: $F_Y(y) = \mathbb{E}[F_Q(y - E)]$. Note that this expectation is taken over the normal variable, $E$. Recall:

$$F_Q(y) = \begin{cases} 0 & y < \alpha \\ \left(\frac{y-\alpha}{\beta-\alpha}\right)^{\gamma/2} & \alpha \leq y \leq \beta \\ 1 & y > \beta \end{cases}$$

Then, using properties of expectations, we have:

$$\begin{aligned} F_Y(y) &= \mathbb{E}[F_Q(y - E)] \\ &= \mathbb{E}_{-\infty}^{y-\beta}[1] + \mathbb{E}_{y-\beta}^{y-\alpha}\left[\left(\frac{(y-E)-\alpha}{\beta-\alpha}\right)^{\gamma/2}\right] + \mathbb{E}_{y-\alpha}^{\infty}[0] \\ &= \Phi\left(\frac{y-\beta}{\sigma}\right) + \mathbb{E}_{y-\beta}^{y-\alpha}\left[\left(\frac{(y-\alpha)-E}{\beta-\alpha}\right)^{\gamma/2}\right] \end{aligned}$$

where $\Phi$ is the standard normal distribution's CDF. Applying the change of variables:

$$V = \frac{(y-\alpha)-E}{\beta-\alpha}$$

and noting that $V$ is normally distributed with mean $(y-\alpha)/(\beta-\alpha)$ and standard deviation $\sigma/(\beta-\alpha)$, we obtain the desired formula:

$$F_Y(y) = \Phi\left(\frac{y-\beta}{\sigma}\right) + \mathbb{E}_0^1\left[V^{\gamma/2}\right], \qquad V \sim \mathcal{N}\left(\frac{y-\alpha}{\beta-\alpha}, \frac{\sigma}{\beta-\alpha}\right) \tag{26}$$

$\square$

**Proposition F.4.** *Let $Y = Q + E$, with $Q \sim \mathcal{Q}_{\min}(\alpha, \beta, \gamma)$ and $E \sim \mathcal{N}(0, \sigma)$. If $f_Y(y)$ is the PDF of $Y$ then:*

$$f_Y(y) = \frac{\gamma}{2(\beta-\alpha)} \mathbb{E}_0^1\left[V^{\frac{\gamma-2}{2}}\right], \qquad V \sim \mathcal{N}\left(\frac{y-\alpha}{\beta-\alpha}, \frac{\sigma}{\beta-\alpha}\right)$$

*Proof.* Let $f_Q(y)$ denote the PDF of $Q$. By the convolution formula for the PDF of a sum we have: $f_Y(y) = \mathbb{E}[f_Q(y - E)]$. Note that this expectation is taken over the normal variable, $E$. Recall:

$$f_Q(y) = \begin{cases} 0 & y < \alpha \\ \frac{\gamma}{2(\beta-\alpha)}\left(\frac{y-\alpha}{\beta-\alpha}\right)^{\frac{\gamma-2}{2}} & \alpha \leq y \leq \beta \\ 0 & y > \beta \end{cases}$$

Then, using properties of expectations, we have:

$$f_Y(y) = \mathbb{E}[f_Q(y - E)]$$

$$= \mathbb{E}_{-\infty}^{y-\beta}[0] + \mathbb{E}_{y-\beta}^{y-\alpha}\left[\frac{\gamma}{2(\beta - \alpha)}\left(\frac{(y - E) - \alpha}{\beta - \alpha}\right)^{\frac{\gamma-2}{2}}\right] + \mathbb{E}_{y-\alpha}^{\infty}[0]$$

$$= \frac{\gamma}{2(\beta - \alpha)}\mathbb{E}_{y-\beta}^{y-\alpha}\left[\left(\frac{(y - \alpha) - E}{\beta - \alpha}\right)^{\frac{\gamma-2}{2}}\right]$$

Applying the change of variables:

$$V = \frac{(y - \alpha) - E}{\beta - \alpha}$$

and noting that $V$ is normally distributed with mean $(y - \alpha)/(\beta - \alpha)$ and standard deviation $\sigma/(\beta - \alpha)$, we obtain the desired formula:

$$f_Y(y) = \frac{\gamma}{2(\beta - \alpha)}\mathbb{E}_0^1\left[V^{\frac{\gamma-2}{2}}\right], \qquad V \sim \mathcal{N}\left(\frac{y - \alpha}{\beta - \alpha}, \frac{\sigma}{\beta - \alpha}\right) \qquad (27)$$

$\square$

