# OpenReview forum: "Hyperparameter Loss Surfaces Are Simple Near their Optima"
_colmweb.org/COLM/2025/Conference — COLM 2025_

### Official Review · Reviewer_DSqh · 2025-04-19

**Rating:** 4
**Confidence:** 2
**Ethics Flag:** 1

**Summary:**

This paper proposes a new theory about hyperparameter tuning, that is, near the optimal point, the loss landscape of the hyperparameters are noisy quadratic, therefore one may apply Taylor approximation or other similar techniques to fit the hyperparameter space, which may help find better hyperparameters.

I think the paper has many problems. I have two main questions:
1. When does the theory hold? Does it hold universally?
2. If the theory holds, is the method proposed in this paper really optimal?

Specifically, I think the theory does not hold in general:
1(a). Generally hyperparameters can be arbitrary, so we are really dealing with a blackbox function. It's hard for me to believe all blackbox functions are noisy quadratic near optimal.
1(b). I think the paper mainly discussed about continuous hyperparameters. Indeed, in the experiment, most parameters they pick are continuous. What if there are discrete parameters? How do you do Taylor expansion and fit it? Or, how is Taylor expansion defined?
1(c). I think the authors mainly discussed some training task like training LLM/Resnet. However, it is not clear to me that, why this will hold for other ML tasks. I am asking about this because years ago, there were a crazy trend about using RL to find hyperparameters, but it turns out the RL algorithm will overfit to the training scenarios easily. Now, the authors did not convince me that this paper is not a similar story: maybe the noisy quadratic case only holds for the parameters they pick on the task they chose, not for other more general cases.

2(a). Even if the theory is indeed true for most cases in AI, I still doubt the noisy quadratic theory is useful. If the landscape is noisy quadratic, it seems that Bayesian methods will be very useful here, because Bayesian methods will assume the function is close to Gaussian, where noisy quadratic is indeed close to Gaussian prior. Therefore, I believe using Bayesian will be as effective. So some comparison is necessary.
2(b). Similarly, since most parameters in the experiment are continuous, I am thinking something like stochastic gradient descent on hyperparameters may help. See e.g., [1] Maybe the authors should compare with that work as well?

Quality: Low. I think the paper is not complete right now. It still has many important discussions missing.
Clarity: Good, the paper is easy to follow.
Originality: Low. I do not think noisy quadratic theory for hyperpameter tuning is specifically novel or useful.
Significance: Low. I think more comparison with existing work, and discussion of generality is requried.


[1] Gradient-based Hyperparameter Optimization through Reversible Learning, Maclaurin et al.

**Reasons To Accept:**

I think the paper cannot be accepted at the current form.

**Reasons To Reject:**

Please check my summary above. I think there are mainly two reasons:
1. The theory does not seem to be general. And the paper did not provide a nice characterization of when this theory holds.
2. If the theory holds, it is not clear to me that the method provided in the paper is the optimal one. I think there are many classical methods can do well, and more comparisons are needed.

---

> ### Author Response · Authors · 2025-06-03
>
> We respectfully disagree with several characterizations in this review.
>
> **"When does the theory hold?"**
>
> Our paper discusses in-depth when the theory holds, and provides extensive empirical evidence for its generality. Formally, Theorem E.2 provides sufficient conditions and a rigorous proof of when the theory holds. Empirically, we show it holds for:
>
> - Diverse architectures (Llama, DeBERTaV3, AlexNet, ResNet, and ConvNext),
> - Multiple tasks (language model pretraining, supervised finetuning, ImageNet pretraining),
> - Both continuous and discrete hyperparameters (batch size, epochs, and blurpool are all discrete),
> - When models are fully trained vs. before convergence (Appendix D).
>
> The asymptotic regime, the region where our theory applies, covers 18-58% of each search space across our experiments. Together, these results show our theory’s generality across architectures, tasks, modalities, training regimes, and different compositions of hyperparameter spaces.
>
> **"mainly discussed about continuous hyperparameters"**
>
> This is incorrect. Our experiments include multiple discrete hyperparameters:
>
> - **Llama**: `warmup_steps` (discrete),
> - **DeBERTaV3**: `batch_size`, `num_epochs` (both discrete),
> - **AlexNet and ConvNext**: `batch_size`, `epochs`, `lr_peak_epoch` (all discrete),
> - **ResNet18**: `batch_size`, `epochs`, `lr_peak_epoch`, `use_blurpool` (all discrete).
>
> In our experiments, discrete hyperparameters did not pose an obstacle to applying our theory. For our proofs, discrete hyperparameters create multiple minima in the loss surface: one for each combination. We discuss this case in Appendix E: Proofs and Theorems. At a high level, you would analyze discrete hyperparameters by doing a separate Taylor expansion for each combination. While this makes the analysis more technically involved, it does not pose an issue in practice&mdash;as our empirical results show.
>
> While these results are in the paper, perhaps they’re easy to miss. In the next revision, we’ll highlight them in a specific discussion of how our theory works for discrete hyperparameters.
>
> **Comparison with Bayesian optimization**
>
> Our work is not proposing an optimization algorithm; we provide tools to understand and quantify uncertainty in hyperparameter experiments. To develop these tools, we demonstrate that loss surfaces have quadratic structure with additive Gaussian noise near their optima (validated in Figures 3-5). This fundamental characterization could inform the design of priors for Bayesian optimization, but that's orthogonal to our contributions. Instead, we use this insight to develop statistical tools for analyzing language models and designing better experiments.
>
> **"theory does not seem to be general"**
>
> All the evidence contradicts this claim:
>
> - The theory is validated on 5 architectures across 3 tasks and 2 modalities.
> - We prove the theory under general regularity conditions in Appendix E.
> - It works even when you use the same search space across different architectures.
>
> On that last point, you state your main concern as: “maybe the noisy quadratic case only holds for the parameters they pick on the task they chose, not for other more general cases”. Appendix D presents an experiment showing precisely that this is not the case. In it, we use the same hyperparameter search space across 3 different architectures and find the noisy quadratic emerges for each one. Because we use the same search space for all three models, it cannot be particularly well-suited to each.
>
> Substantial evidence and carefully designed experiments all demonstrate that our theory holds across many common scenarios.

---

### Official Review · Reviewer_rHdb · 2025-05-10

**Rating:** 4
**Confidence:** 2
**Ethics Flag:** 1

**Summary:**

The paper introduces a noisy‑quadratic parametric model that captures how random search improves validation loss once hyper‑parameters are near the optimum. Experiments on three deep‑learning setups show that this simple model fits the tail distribution well and extrapolates tuning‑curve performance.

**Questions To Authors:**

See Reasons to reject.

**Reasons To Accept:**

- The paper proposed a principled model for tuning‑curve forecasts with random search, with support of empirical validations.
- The paper is well structured and clearly written.

**Reasons To Reject:**

In order to confirm the paper’s findings, additional experiments should be conducted to cover more diverse search spaces. Because the conference focuses on language modeling, extensive evaluations with recent language models of various sizes and types would be more appreciated.

The paper investigates only random search and claims that the proposed model can be used to build tools for analysis. However, because the analysis is limited to random search, other common HPO methods (e.g., Bayesian optimization, BOHB [1]) cannot benefit from it.

[1] Falkner, Stefan, Aaron Klein, and Frank Hutter. "BOHB: Robust and efficient hyperparameter optimization at scale." International conference on machine learning. PMLR, 2018.

---

> ### Author Response · Authors · 2025-06-03
>
> Thank you for noting our "principled model" and clear presentation.
>
> **"experiments should be conducted to cover more diverse search spaces"**
>
> Our experiments already include substantial diversity:
>
> - **Language models**: Llama 33M (1,024 runs) and DeBERTaV3 (1,024 runs),
> - **Vision models**: ResNet18 (1,024 runs), AlexNet (512 runs), and ConvNext (512 runs),
> - **Tasks**: Language model pretraining, ImageNet pretraining, and NLI finetuning,
> - **Datasets**: SlimPajama-6B, MultiNLI, and ImageNet.
>
> The main text covers scenarios from language and vision, including one of the most popular LLM architectures. In addition, Appendix D shows how results remain stable as you vary the architecture while keeping the hyperparameter search space constant. The theory consistently holds across all settings, with the asymptotic regime covering 18-58% of each search space.
>
> While more experiments are always helpful, we trained over 3,000 models for this work and the current experiments demonstrate that our theory generalizes across architectures, tasks, and modalities.
>
> **"other common HPO methods… cannot benefit from it."**
>
> The goal of our work is not to benefit HPO methods, but to study the hyperparameter loss surface and develop tools for language modeling experiments. In development, researchers run many experiments where training the best model is not the goal. Instead, these experiments have goals like characterizing scaling laws or determining which data mix is better. HPO methods are not designed for these sorts of experiments.
>
> Our theory and the tools we develop characterize the loss surface itself. Random search provides a powerful tool for *analyzing* the loss surface because of its clean mathematical properties; however, the structure it discovers exists regardless of the search algorithm used. This structure could benefit adaptive HPO methods by informing their design or configuration, but that is future work.

---

> > ### Comment · Reviewer_rHdb · 2025-06-09
> > **Reply**
> >
> > I would like to thank the authors for their responses. I decide to keep the score.

---

### Official Review · Reviewer_tyYN · 2025-05-11

**Rating:** 7
**Confidence:** 1
**Ethics Flag:** 1

**Summary:**

This paper describes experimental and theoretical results pertaining to hyperparameter optimization. They find that the loss curve is well approximated by a quadratic in the vicinity of the maximum value. Based on these results, they propose methods for estimating confidence intervals for the loss and a "scaling law" for predicting the best possible loss on the basis of relatively few (possibly very expensive) steps of random search.

This is (in my opinion) a fairly small result with limited applications, and it's not surprising to anyone with experience in training neural nets. However, their empirical evaluation and theoretical framing put that intuition on a more solid footing and may be something that later work can build on.

**Reasons To Accept:**

Sensitivity to hyperparameter settings is an ongoing challenge for evaluating large neural nets. Anything that can help quantify the variation introduced by hyperparameter search is potentially very useful.

**Reasons To Reject:**

A small point: I'm not sure I'd describe the proposal here as a scaling law, since there's not really any actual scaling involved. Maybe "quadratic bound" or "predicted maximum" or something like that would be a better term for it?

More generally: there's a gigantic literature on hyperparameter optimization and loss surfaces in deep learning context and in non-local optimization in general. It would be beyond the scope of a paper like this to try to survey it, but it would be helpful if the connections and overlap with tthose other results in the "related work" section.

---

> ### Author Response · Authors · 2025-06-03
>
> Thank you for recognizing the value in our work’s ability to "quantify the variation introduced by hyperparameter search."  We appreciate your constructive feedback.
>
> **"scaling law" terminology**
>
> We refer to our result as a scaling law because it extrapolates how performance improves as you scale the iterations of random search, similarly to existing scaling laws which predict how performance improves as you scale data or compute. However, if you still object to this terminology, we’d be happy to consider alternatives.
>
> **Related work on hyperparameter optimization**
>
> Thank you for the suggestion. We'll add discussion of HPO methods to clarify how our tools for understanding the loss surface might inform their design. However, we emphasize our focus is on *understanding* loss surfaces, not optimizing them. Work on Bayesian optimization and other HPO methods is orthogonal to our own&mdash;typically they make no assumptions about the hyperparameter loss surface, while we reveal its mathematical structure.
>
> As for work on the hyperparameter loss surface, our discovery of quadratic structure differs fundamentally from prior findings of unimodal/convex (Pushak & Hoos, 2022) or fractal (Sohl-Dickstein, 2024) surfaces, as already mentioned in the related work.

---

> > ### Comment · Reviewer_tyYN · 2025-06-08
> > **Followup**
> >
> > I thank the authors for their response.
> >
> > I still don't agree that this should be called a scaling law. Chinchilla-style scaling laws are mostly about data and model size and how they trade off with compute. As far as I can tell, there's nothing in your results that would tell you in advance the optimum quantity of FLOPs to spend on hyperparameter optimization for a model and training set of a given sizes. What you're doing seems to me to be a lot more in the tradition of optimal stopping or PAC learning. But that's kind of a minor point of terminology.
> >
> > About the literature review, I take the authors' point. I'll change the score.
> >
> > But: if there is another version of this paper, I would encourage to authors to try to make it clearer in the paper what they're actually doing. It took me four readings to figure it out and from looking at the discussion I don't think some of the other reviewers ever got it. That's because this paper seems to be about one thing (which is a very active area of research and we've all read tons of papers about) but it's actually about something completely different (which no one else is doing and we've never seen before). The fact that it confused all the reviewers so much is a sign that it could be making its point in a better way.

---

> > > ### Author Response · Authors · 2025-06-10
> > >
> > > Thank you for your thoughtful feedback and **thank you for taking the time to really understand our work** and its contributions.
> > >
> > > Your comment is insightful and captures the challenge in presenting this work: we discuss a familiar topic (hyperparameters) but from an unfamiliar angle: managing uncertainty about hyperparameters when doing experiments. As you point out, it's related to hyperparameter optimization ("a very active area of research... we've all read tons of papers about") but at the same time it tackles an important issue that "no one else is doing and we've never seen before".
> > >
> > > You're right, and we will adjust the introduction to clarify that we tackle a new problem. *We ask that the other reviewers consider your point as well:* **a new research direction, even an important one, can be difficult to discuss when it's unfamiliar.** In our work, we aim to contribute some key tools and develop the vocabulary to make progress on this problem&mdash;especially because LLMs have made it so necessary to have better tools for our experiments.

---

### Official Review · Reviewer_rVAh · 2025-05-13

**Rating:** 4
**Confidence:** 2
**Ethics Flag:** 1

**Summary:**

This submission seems to propose an analytical model of how the best (validation) score for a model is distributed in the neighbourhood of the optimal hyperparameters. Experimental results seem to produce reasonably accurate fit to the theoretical predictions.

The main concern with the manuscript is that it is very unclear regarding not only what is done, but even what the actual goal is. The starting point is a study of the hyperparameter loss surface. In a typical learning setup, model parameters are estimated by fitting parameters for a given set of hyperparameters, the suitability of which is assessed by computing some empirical estimate of the expected risk, such as a validation loss, hence the "hyperparameter loss surface", or high-dimensional surface of the validation loss over the space spanned by the hyperparameters. (If this is not what is intended here, then a central point has been completely lost on this reviewer.) Yet most of the remainder of the paper seems concerned with characterizing the (one-dimensional) distribution of the loss values near their extremum, rather than the high-dimensional loss surface.

The setup relies on (i.i.d.) random hyperparameter search, which provides a convenient theoretical framework for deriving extremum distributions. However this does not seem a very realistic setting for h.p. tuning, where one would typically expect prior hyperparameter scores to inform h.p. sampling, breaking the i.i.d. assumption. Similarly to "random search", many concepts and techniques are not described and not referenced, and left for the reader to figure out. To give one example: the "censored maximum spacing estimation" used to fit one of the distributions (Fig. 3, caption).

This may be novel and original work, but the significance is entirely unclear. It would of course be highly relevant and useful if it provided an improved method for tuning hyperparameters, or at least help produce better hyperparameters. This seems to be what Section 4.3 is promising; unfortunately that section shows more comparison between empirical estimates and theoretical predictions.

**Questions To Authors:**

l. 129, "retrain the same hyperparameters" : presumably you mean retrain the model using the same hyperparameters?

**Reasons To Accept:**

This submission seems reasonably successful at modelling whatever it is they are attempting to model.

**Reasons To Reject:**

Although this submission is well written overall, the main body of the paper lacks clarity and specificity to the point that it is unclear what is done exactly, and whether and how this can have any practical use.

---

> ### Author Response · Authors · 2025-06-03
>
> Thank you for your detailed feedback. We apologize for the clarity issues and will address them in the next revision.
>
> **"unclear regarding not only what is done, but even what the actual goal is"**
>
> Our goal is stated in the abstract: to "discover novel structure in the hyperparameter loss surface and propose a new theory describing it". We accomplish this, empirically showing that near the optimum the loss surface is approximately quadratic with additive Gaussian noise. With this insight, we develop a theory that yields better statistical tools for understanding hyperparameter loss surfaces and the language models they come from. These tools include:
>
> - Confidence intervals for the best achievable loss,
> - Estimates for the hyperparameters’ effective dimensionality,
> - Extrapolation of random search’s convergence.
>
> Note that we do **not** try to build a better hyperparameter optimization algorithm. Instead, we seek to discover structure and develop better tools for language modeling experiments.
>
> **"(one-dimensional) distribution... rather than the high-dimensional loss surface"**
>
> The noisy quadratic distribution, $\mathcal{Q}(\alpha, \beta, \gamma, \sigma)$, directly encodes the high-dimensional surface’s properties:
>
> - $\gamma$: the intrinsic dimension (the effective number of hyperparameters),
> - $\beta$: the surface’s maximum (the best achievable score),
> - $\sigma$: variability around the surface (the noise due to random seeds),
> - $\alpha$: the size of the asymptotic regime (how tight the search space is).
>
> It is impossible to directly visualize the loss surface in high dimensions. That’s why our theory is so helpful: it makes it possible to study the loss surface by creating a direct link between it and the noisy quadratic distribution.
>
> For example, we find $\gamma = 2$ across three different architectures when we use the same search space (Figure 7). Despite this search space having 7 nominal hyperparameters, only a 2 dimensional subspace matters near optimum. This dimension holds constant across all three architectures, suggesting that it’s *a robust, intrinsic property of that hyperparameter loss surface*.
>
> **“random search… does not seem a very realistic setting for h.p. tuning”**
>
> Our interest is not in hyperparameter tuning, but designing better tools for language modeling experiments. As you point out, random search provides a convenient theoretical framework for understanding the loss surface. We do not seek the most efficient hyperparameter tuning algorithm, and random search certainly isn’t it. Instead, it’s reasonably effective, easy to implement, and offers an important standard when comparing models across papers (where researchers need a single standard to enable fair comparisons). Since we are not interested in tuning hyperparameters, but rather comparing models, random search is the best starting point.
>
> Of course, when practitioners tune these models for production, they’ll want to use the most efficient algorithms. In that case, our theory reveals properties of the hyperparameter loss surface that are independent of the search algorithm used. Adaptive methods can be designed to exploit these properties, once discovered.
>
> **"censored maximum spacing estimation"**
>
> This is a standard statistical method for fitting distributions (Cheng & Amin, 1983; Ranneby, 1984), and describing it would be out of scope for our paper. Since it is less widely known than maximum likelihood, we do in fact provide citations for it and the other methods we use when we describe them in Section 3: *Experimental Setup* (e.g., lines 161 or 169).
>
> **"novel and original work, but the significance is entirely unclear"**
>
> Our theory offers new statistical tools for studying language models. These include:
>
> - Confidence intervals for the best achievable performance,
> - An estimator for the effective number of hyperparameters,
> - Extrapolation of hyperparameter tuning curves as a way to compare models.
>
> Section 4.3 demonstrates this practical significance by showing dramatic improvements in confidence bands for tuning curves&mdash;a tool for comparing models while accounting for hyperparameter tuning difficulty. The current nonparametric bands become trivial after 8 iterations (Figure 6), while our parametric bands extrapolate past all 48 iterations used to construct them, enabling uncertainty quantification when interpreting experimental results.

---

> > ### Comment · Reviewer_rVAh · 2025-06-03
> >
> > Thanks to the author for their substantial replies. They do provide additional context to better grasp their goal and some of the methods used.
> >
> > After reading the reviews and corresponding replies, there is still some struggle understanding how the described methods address the stated goal, and what the real-life impact is. With a goal of studying the hyper-parameter loss surface and develop *tools* for language modeling experiments, it would be highly relevant to have an idea of what type of LM experiment do benefit from these tools, and how. What is the reader able to do after reading this paper, that they could not do before?
> > [E.g., is it useful for contrasting models? This is not apparent from Figs. 3, 6 where different model types and tasks are shown.]
> >
> > I will absolutely acknowledge that I may not have the proper context or background to fully appreciate the significance of this work, and have updated the confidence score to reflect this. I would also respectfully encourage the authors to clarify their methods, their use and the impact of this work in order to make its benefits more apparent to the reader.

---

> > > ### Author Response · Authors · 2025-06-10
> > >
> > > Thank you for your careful consideration of our rebuttal. You ask an important question.
> > >
> > > > What is the reader able to do after reading this paper, that they could not do before?
> > >
> > > The reader gets several new tools from our paper.
> > >
> > > **Confidence intervals for the best achievable performance.** Our work gives confidence intervals for the validation score that a model would get with perfectly tuned hyperparameters. These intervals remove the guess work around whether or not you tuned the hyperparameters enough. You could use them *to compare models* or *to decide if it's worthwhile to look for better hyperparameters*. To the best of our knowledge, our work provides the first and only method for doing this.
> > >
> > > **An estimator for the effective number of hyperparameters.** The effective number of hyperparameters is a fundamental property of a hyperparameter search space. It strongly influences how difficult a model is to tune because it represents the intrinsic dimension of the hyperparameter loss surface. Sometimes a new hyperparameter does *not* increase this effective number; other times a new hyperparameter increases it and makes hyperparameter tuning more difficult. Researchers could use our estimator *to determine if adding a new hyperparameter is worth the complexity it brings* or *to test whether a new hyperparameter is related to existing ones*.
> > >
> > > **Confidence bands for tuning curves.** Tuning curves (the validation score as a function of tuning iterations) are an established tool from the literature for comparing models while accounting for hyperparameters [1, 2, 3]. Our work provides the first confidence bands that can *extrapolate* tuning curves. This extrapolation allows you to compare models while accounting for hyperparameter tuning effort beyond the budget you actually used to tune the models.
> > >
> > > These tools enable you to quantify your uncertainty with respect to hyperparameters and thus to design more informative experiments. Our paper presents these tools, but it doesn't provide a tutorial on how to use them. You give an excellent suggestion, and we will add a discussion of how to use these tools in order to clarify the use of our methods and their impact.
> > >
> > > [1]: [*Show Your Work: Improved Reporting of Experimental Results.* (Dodge et al., 2019)](https://aclanthology.org/D19-1224/)
> > >
> > > [2]: [*Expected Validation Performance and Estimation of a Random Variable’s Maximum.* (Dodge et al., 2021)](https://aclanthology.org/2021.findings-emnlp.342/)
> > >
> > > [3]: [*Show Your Work with Confidence: Confidence Bands for Tuning Curves.* (Lourie et al., 2024)](https://aclanthology.org/2024.naacl-long.189/)

---

### Author Response · Authors · 2025-06-03
**Overall Response**

We thank the reviewers for their thoughtful engagement with our work. We believe there has been a fundamental misunderstanding about our paper's goals and contributions. Our work is **not** about creating a better hyperparameter optimization algorithm, but rather about **understanding the mathematical structure of hyperparameter loss surfaces** and providing **statistical tools for language modeling experiments**. This distinction is crucial and addresses most of the concerns raised.

Our key contributions are:

- **A theory of the hyperparameter loss surface** built on an assumption that we empirically validate: it is quadratic with additive Gaussian noise near the optimum,
- **New statistical tools** for estimating its properties like the *effective number of hyperparameters* and *confidence intervals for the best achievable performance*,
- **Extensive empirical validation** across 2 modalities, 3 tasks, and 5 architectures showing this structure consistently emerges in practice.

Our work does not aim to speed up hyperparameter search, but rather to enable better experiments for language model development. It facilitates reliable comparisons across papers by providing a principled foundation for quantifying uncertainty with respect to hyperparameters. Random search serves purely as a standard for comparing models, and it is well-suited to this role because it is a strong baseline, easy to implement, and always available. The uncertainty quantification we provide is an essential, missing piece for rigorous experimental comparisons.

Finally, we point out that while different reviewers have strong reactions to our work, they actually disagree. Reviewer tyYN states our theory is "not surprising to anyone with experience in training neural nets," while Reviewer DSqh claims "the theory does not hold in general." Of course, these cannot both be true and in particular they underscore exactly why our empirical and theoretical study is so necessary. *The community does not know the answers to these questions.* We take a simple premise&mdash;hyperparameter loss surfaces are quadratic with normal noise near their optima&mdash;and we bring it to its logical conclusion. In doing so, we make significant technical contributions: proving *a novel limit theorem*, deriving *a new family of probability distributions*, and *extensively validating* that they describe the hyperparameters of real-world models.

The community deserves a conclusive answer to these questions and the tools that they provide for language modeling experiments.

---

### Decision · Program_Chairs · 2025-07-08

**Decision:**

Accept

**Comment:**

This paper presents a theory about the structure of hyperparameter loss surfaces near their optima, proposing that they follow a noisy quadratic distribution. The authors validate this across multiple architectures and tasks, and develop statistical tools for uncertainty quantification in hyperparameter experiments.

The reviews split in terms of enthysiasm especially wrt understanding the paper's contributions. Reviewer tyYN (score: 7) appreciated that this work is about understanding loss surface structure for experimental design, not proposing a new opt algorithm. They note it's "a fairly small result" but one that "put[s] that intuition on a more solid footing." In contrast, three other reviewers (scores: 4, 4, 4) appreciated that this was not the main goal of the paper, critiquing it for not offering a better (and more cocnrete) hyperparameter/optimization method and that it is incomplete (i.e., lacking enough empirical evidence to support some of the hypotheses)

Strenghts:

New (according to some revs) theoretical contribution characterizing hyperparameter loss surfaces as noisy quadratic near optima
Solid empirical validation across 3,000+ training runs spanning multiple architectures, modalities, and tasks
Provides practical guidance for uncertainty quantification
Well-written
Weaknesses:

The paper fails to clearly communicate its goals early on, leading to widespread reviewer confusion
Limited discussion of when the theory might break down. Some reviewers are unconvinced about generalizability across arbitrary setups
Unclear practical impact
Some terminology choices (e.g., "scaling law") may be misleading
The core issue here is not the technical merit but perhaps communication. As tyYN notes: "this paper seems to be about one thing... but it's actually about something completely different... The fact that it confused all the reviewers so much is a sign that it could be making its point in a better way."

The authors' rebuttal clarifies to a degree their actual contributions, but it's unclear if the reviewers agree even in light of it. This is challenging work to evaluate because it addresses an important but perhaps not super familiar problem. The empirical work is thorough, the theory appears interesting and sound, and the tools could be useful for the community. However, it is *extremely critical* that the authors make their goals extremely clear from the beginning, taking into account the reviews and the meta-review, and significantly revise the presentation.